# Gamification with Scratch or App Inventor in Higher Education: A Systematic Review

David Pérez-Jorge *[ID] and María Carmen Martínez-Murciano [ID]

Didactic and Educational Research, University of La Laguna, 38200 La Laguna, Spain
* Correspondence: dpjorge@ull.edu.es

**Abstract:** Programming skills should be taught and developed; Scratch and App Inventor are two tools that can contribute significantly to developing this competence in university students. This study aims to investigate the use and effect of the programming language Scratch and App Inventor on the development of skills and competencies for learning (autonomy, attention, motivation, critical thinking, creative thinking, computational thinking, communication, problem solving and social interaction) in higher education. To achieve this goal, a systematic review of articles in English and Spanish was carried out using the PRISMA statement (research publication guidelines designed to improve the integrity of systematic review and meta-analysis reports). A search for studies was conducted in the Web of Science (WOS), Dialnet, and SCOPUS. A total of 405 papers were analyzed, of which 11 were finally selected. The results showed that both Scratch and App Inventor favor the development of skills and competencies for learning in the context of higher education, despite being underutilized strategies that all knowledge disciplines should promote.

**Keywords:** App inventor; scratch; higher education; motivation; creative thinking

## 1. Introduction

This systematic review study focused on the advances and educational application of Scratch and App Inventor tools in the context of higher education. The analysis of the innovation experiences developed and applied in different universities will allow us to assess the results of the development of generic skills and competencies, such as autonomy, attention, motivation, critical thinking, creative thinking, computational thinking, communication, problem solving, and social interaction. In the 1990s, educational innovation came together with the Internet; advances in programming and the design of applications and programs for learning have placed artificial intelligence (AI) at the center of the development of tools for intelligent programming, whose implication in the models and forms of teaching currently have a great impact. It is used in educational software for data analytics, performance evaluation, learning analysis of dynamic curricula, and Big Data, among other applications.

### 1.1. Programming in Gamification

Computer programming is taught in different areas and subjects; as stated by [1], the importance of achieving new skills and having both theoretical and practical knowledge of programming has been understood. The basis of learning programming language is in the realization of projects focused on programming. This methodology of work with highly practical content focuses on processes and feedback and an effective evaluation of progress and difficulties in programming, involves the development of programming skills that exceed the final result of the projects, and it is necessary to perform a procedural evaluation of the projects and not only focus on the evaluation of the results obtained by the students. If we combine ubiquitous learning and motivation, programming through gamification is a fun way to learn. Gamification is gaining momentum as an active methodology in

education. Children learn by playing from the beginning of their lives. Nowadays, the motivational power of gamification extends to higher-education contexts, where, as in other educational stages, students show interest in this learning [2,3].

## 1.2. Appropriateness of Programming Languages

The idea of the new "lightweight society" is that it seems that everything changes and that before learning something, it may already be out of use, and it has been thought that this could happen in the case of the Scratch and App Inventor. The results on the benefits of learning them support their use, however; for example, the authors of [4] created the FORmula TRANslation or FORTRAN programming language in 1954, which is still current despite its age. At the time, much of the effort of its creators was focused on overcoming the deficiencies of the computers of the time. Nowadays, programming languages help in the learning process; their simple and intuitive use favors their use from an early age. An appropriate way to encourage student motivation could involve encouraging students to create their own applications (apps), developed by App Inventor, or programming with Scratch in the classroom or outside of it.

## 1.3. Conceptualization of Scratch

Nowadays, students can learn through AI. The Scratch project was created by the Lifelong Kindergarten research group of the MIT Media Lab [5]. Based on the development of a creative workshop to achieve a change in student learning, it involves designing and creating interactive stories and animations, expressing oneself, and using creative thinking through technology and learning to program.

Scratch is the most widely used programming language in the classroom because it can be used from early childhood to higher education. The stage where most programming is performed with this language corresponds to the university stage, followed by the secondary-education stage [6]. Scratch is a programming community that promotes computational thinking and the skills of conflict resolution, creative teaching and learning, self-expression and collaborative work, and equality. It was designed for children between the ages of 8 and 16 to learn through creation and exploration, but it is now used by virtually everyone, regardless of age. The designers advise Scratch users to create ideas and try them out, and if they do not work for them, to try again and make changes. This programming language is used in more than 200 countries and has been translated into 70 languages. Students learn with Scratch at all educational levels and all levels of education through different disciplines such as art, computer science, language, mathematics, and social sciences [7].

## 1.4. Conceptualization of App Inventor

App Inventor is also an educational resource, a visual programming environment using blocks created in 2010. Students from 4 years old to higher education use it to create mobile applications, support work, and games. According to [8], App Inventor enables users to create Android apps such as program data input. Mobile applications could be created for automatic process measurement, systems, and robot control. In this sense, [9] considered that students who create apps in the classroom achieve motivating, globalized, constructive, meaningful, technological, and competent learning. Digital production can awaken vocations towards professional careers oriented to engineering and the scientific and technological world. App Inventor can be used in any educational context; it should be given a prominent place among digital production tools and used from the age of three and at all educational stages. Other apps that students have developed with App Inventor contribute to social welfare and gender equality, such as Impegno Dogma, which gives support to women victims of gender violence; they also contribute to sustainable development, such as the Fern app, which helps people to track their carbon footprint; further, apps can promote diversity and acceptance, such as the app UniP, which was created by a teacher to

show the beauty of diversity and shed light on the autism community [8]. Moreover, App Inventor could use learning analytics to enhance performance.

An interesting case was proposed by [10]; it showed that two 10-year-old boys in India developed the Plug the Drip app, which contributed to environmental protection by avoiding water wastage. This app monitors taps to detect leaks or drips and sends a notification to the user; it even provides a list and map of plumbers. It also shows how two 12-year-old teenagers in Pakistan invented the Lighting Automation System app, significantly improving the lives of dependent people with reduced mobility or needing medical rest because they automated home lighting and could turn the lights in the house on and off without the need for a caregiver. A university student from Brazil created the Meteor ID application that helps users to know if a certain rock could be a meteorite from outer space; the application answers questions about the characteristics of the sighting in a test, and the cases identified as positive have the photos report by mail or to social networks of the Meteoritos do Brasil project, which seeks to identify new meteorites. A retired Japanese professor of computer science, Fujio Yamamoto, created the app Q-Learning On Your Palm to explain the basic idea and mechanism of reinforcement learning (especially Q-Learning), which is a field of artificial intelligence in which a robot is trained to move in different directions.

The evidence and Ies evaluated on the use of programming applications have shown positive results in relation to competencies, and the scope of the development of these competencies is the focus of this review study.

We aimed to investigate the Scratch programming language and the App Inventor's uses and effects on developing skills and competencies (autonomy, attention, motivation, critical thinking, creative thinking, computational thinking, communication, problem solving, and social interaction) in university education.

We also aimed to determine the formative contexts in which the visual programming language Scratch and the App Inventor are promoted or used in university education.

From these general objectives, the following specific objectives are proposed.

− To determine the types of thoughts and skills (autonomy, attention, motivation, critical thinking, creative thinking, computational thinking, communication, problem solving and social interaction) that Scratch and App Inventor develop most in university students.
− To assess the interest that the use of computational programming through Scratch and App Inventor arouses in research.

## 2. Materials and Methods

Programs that promote learning in schools tend to focus on different areas; therefore, we searched for studies that addressed specific areas of Scratch and App Inventor and not only those that talked about learning generically. In addition, the studies considered were only estimated for those involving students and teachers, not considering those developed by other potential users (principals, families, etc.). The studies on these programs should have provided benefits and/or limitations in their implementation in relation to the acquisition of programming knowledge and the development of competencies related to computational thinking. Therefore, at the beginning of this study, criteria were established for the inclusion and exclusion of documents to carry out an adequate selection of sources (see Table 1).

**Table 1.** Inclusion and exclusion criteria considered.

| Inclusion Criteria | Exclusion Criteria |
|---|---|
| - Studies in English and Spanish<br>- Papers written within the last 5 years<br>- Papers focused on university education<br>- Research papers<br>- Studies on App Inventor and/or Scratch<br>- Open access articles | - Studies in languages other than English and Spanish<br>- Documents older than 5 years<br>- Documents from other educational stages.<br>- Reflection articles<br>- Articles that do not base their results on App Inventor and/or Scratch. |

### 2.1. Type of Study

The methodology used in this study was of a mixed and interpretative nature, through a systematic review of the scientific literature on the research topic, following the PRISMA (preferred reporting items for systematic reviews and meta-analyses) statement. Databases, articles, and books on scientific production developed in relation to the educational use of the visual programming language and App Inventor were analyzed. The previous existence of Scratch and App Inventor studies on the subject justifies the review presented in this paper [11]; through this study we intend to show in a synthetic way the results and/or the state of the art on the topic.

The PRISMA statement is a guide on the conceptual and methodological aspects considered during the development of systematic review studies [12]. It consists of accessing scientific documentation on a topic to conclude objective reasoning through evidence based on it [13]. It aims to begin new lines of research based on analyzing research findings up to a given moment. Specifically, we aimed to determine how Scratch and App Inventor are applied in universities and their effects on student learning. The Prism 2020 statement [14] was updated to take advantage of the benefits of technological innovations such as natural language processing and new terminology.

The research question that guided the research process was "do studies on the use of the Scratch programming language and the App Inventor show effects on the learning of programming and competence development in university students?"

### 2.2. Review

To ensure that the collection and purging of information was rigorous and specific, the topics were extracted from the keywords of the research question. To refine the search for topics, an initial process was carried out to identify the keywords most frequently used in the studies that addressed this line of work, analyzing their relevance and appropriateness to the objective of our study. This process is specified in point 2.4, as it is considered useful for guiding future research work in this field. In the beginning, the search strategy was very general and showed too many documents that did not represent or fit the objective of this study.

The Boolean terms OR/AND were used as Booleans; specifically, the combinations used for the search were Scratch OR App Inventor AND Education. The first terms were specifically searched for, and the different Boolean markers were combined with the different stages of formal education to determine the preponderance in relation to the different educational stages. University education was definitively established as the stage of interest for the study, mainly because it is the stage in which most studies on the subject have been carried out. The combination for the final search was Scratch OR App Inventor AND College. For the search in Spanish, the topics used were Scratch OR App Inventor AND Universidad.

### 2.3. Resources

The resources used for the information search strategy during the study were three electronic databases from the search engine of the University of La Laguna library, from

which documents from the Web of Science (WOS), Scopus, and Dialnet were accessed. These databases were the main ones that house research in the educational field.

### 2.4. Procedure

An exhaustive search was started in these databases using the keywords, after which the inclusion criteria were applied, eliminating those documents that did not meet them, and using the Mendeley bibliography manager, duplicate documents were eliminated. After this, we read the documents' titles and abstracts, selecting the most appropriate papers for analysis. From the selected texts, a complete reading was carried out, and the final decision was made on the selection or rejection of documents.

The information extracted was obtained through content analysis of the ten studies based on previously established criteria relevant to this study. The analysis refers to case studies described in research conducted on the main features and formative effects of Scratch and App Inventor in university classrooms. The initial starting selection for the analysis was as follows:

A total of 204 papers were found in Scopus for the combination (Scratch OR "App Inventor") AND College.

In Dialnet, 12 documents were found for (Scratch OR "App Inventor") AND "Higher Education".

In WOS, 189 results were found for (Scratch OR "App Inventor") AND College.

A total of 405 results were obtained. Titles and abstracts were read applying inclusion and exclusion criteria, and 347 articles were removed that were either written prior to 2017 (N = 220), in a language other than English or Spanish (N = 23) or were not open access (N = 104). Following this, duplicates were removed, excluding a total of 17 articles. This allowed the selection of 41 papers that were read on a full-text basis. Eleven papers were selected as suitable for the study.

The detailed search is shown below. See Figure 1.

### 2.5. Characteristics of the Included Studies

The eleven articles selected for the review were research studies published in English (N = 10) and Spanish (N = 1) between the years 2016 and 2022, which ensured updated results on the application of Scratch and App inventor programs in universities. The studies were conducted in Colombia, China, Ecuador, Egypt, England, Japan, Nigeria, Spain, and the United States.

Six studies used quantitative methodology (54.55%; N = 6), two were mixed studies (18.18%; N = 2), one was qualitative (9.09%; N = 1), one was project-based learning (9.09%; N = 1), and one was real-life testing (9.09%; N = 1).

Some assessment instruments overlapped across the studies, with 5 being used in the 11 items selected. These were questionnaires (44.44%), surveys (27.77%), observation (11.11%), objective tests (11.11%), and interviews (5.55%).

As seen in Figure 2, the most frequently used instruments were questionnaires, followed by surveys, and the least used were interviews. In all the studies, the selected sample consisted of university students; only one study also included teachers. As for the results, it should be noted that most of the programs met the objectives they had set; only one found a significant sample of students who considered that the program was not fast in its interaction and did not find precision in the personalized exercise recommendations. See Figure 2.

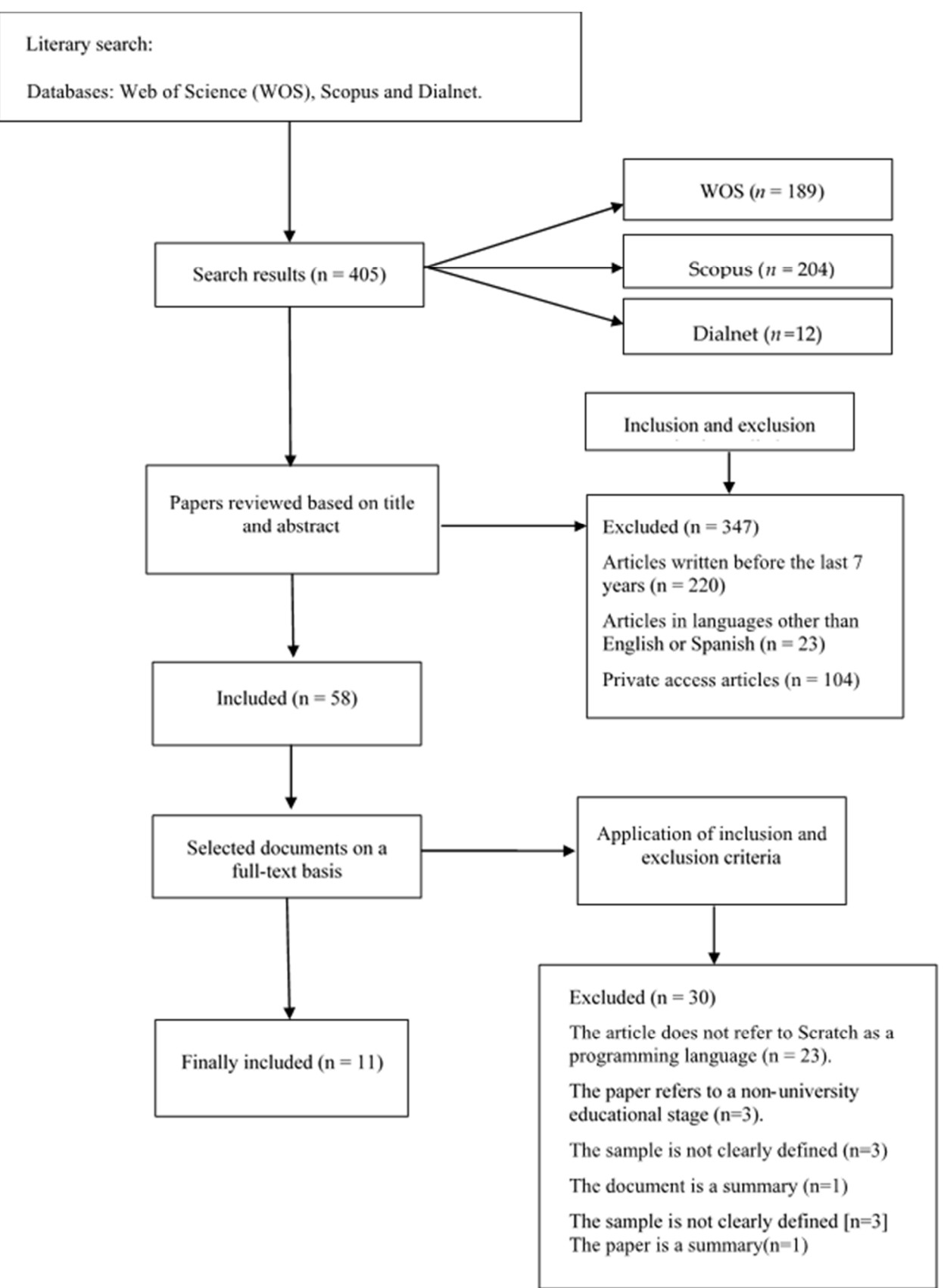

**Figure 1.** Document search flowchart.

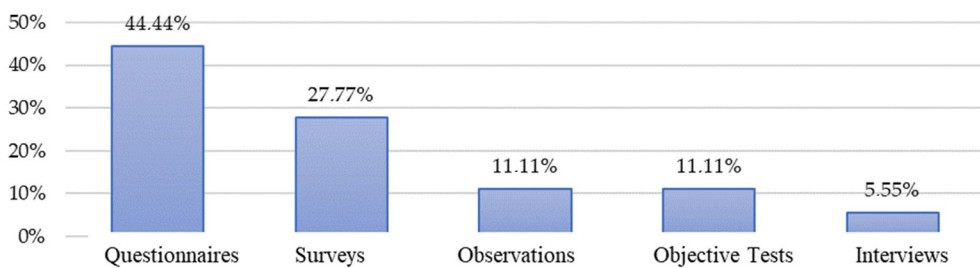

**Figure 2.** Instruments in selected studies.

Nine studies used the Scratch language, and two used App Inventor. All the information can be seen in Table 2, specifying the authorship, year of publication, purpose, sample, country, methodology with evaluation instrument, and main results.

**Table 2.** Explanatory table of consultation documents.

| Author and Year | Objectives | Sample | Country | Method | Results |
|---|---|---|---|---|---|
| [15] Areed et al. (2021) | To examine the impact a gamified app (App Inventor) can have on students' specific learning performance in mathematics and how to integrate gamified electronic quizzes into a mobile app as a learning tool. | 60 s-level engineering students from the University of Damietta. | Egypt. | Quantitative method comparing paper questionaries to gamified electronic quizzes. | Students who used gamified electronic quizzes felt that they improved their critical thinking and demonstrated more learning skills compared to students who used traditional paper-based quizzes. The authors encourage teachers to use gamification as a modern, innovation-oriented tool through which students can participate in an engaging and competitive experience. |
| [16] Campbell and Atagana (2022). | To use Scratch in learning constructionist programming to examine the nature of student engagement in a higher-education classroom. | 96 students (a full 1st-grade computer science class). | Nigeria. | Quantitative and descriptive method with structured observation intervention in-class sessions. The tools are questionnaires. | Using Scratch motivates and piques the interest of novice students in programming, where direct exposure to text-based programming might hinder their interest. |
| [17] Saez-López et al. (2016) | To analyze the interactions, attitudes, and practices of students who participate in dynamics, multimedia, and cross-cultural and intercultural activities and if Scratch supports the acquisition of basic programming concepts. | 113 students from 3 universities. | Mexico, Japan, and Spain. | Quasi-experimental mixed method with a DBR approach (design-based research) instruments: questionnaire, test, and surveys. | The students favor using Scratch and emphasize that it is intuitive, easy to use, fun, and perfect for presentations and animations. It makes them understand the use of multimedia content through block programming. |
| [18] Cárdenas-Cobo et al. (2018) | To improve the learning of computer programming in students using Scratch + an RS web application for exercises (problem statements) to achieve their motivation and less teacher intervention. | 64 computer science and industrial engineering students. | Ecuador. | Real-life testing methodology. Satisfaction questionnaire instruments. | 60% percent of the student body of both careers agree that the proposed system is better, and 40% strongly agree. |

**Table 2.** *Cont.*

| Author and Year | Objectives | Sample | Country | Method | Results |
|---|---|---|---|---|---|
| [19] Cárdenas-Cobo et al. (2020) | To learn if using Scratch together with its extension caramba improves motivation and satisfaction by offering more individualized programming in programming fundamentals classes in computer science and industrial engineering. | 64 computer science and industrial engineering students. | Ecuador. | Quantitative methodology with questionnaires. | More motivated students, grades improved by 8% compared to using Scratch alone and by 21% compared to teaching without Scratch. |
| [20] Cárdenas-Cobo et al. (2021) | To use Scratch in the Fundamentals of Programming classes to reduce school failure. | 74 students in 1st year of Computer Systems. | Ecuador. | Quantitative descriptive with a quasi-experimental approach. Instruments: questionnaires. | Scratch was shown to improve programming grades by a factor of 4. Students were motivated to continue learning with Scratch. |
| [21] Shen and Qi (2020) | To achieve autonomous learning in students with psychological problems by integrating Scratch programming software to improve skills. | 81 psychology students. | China. | Quantitative methodology such as questionnaires and survey instruments. | Students who used Scratch scored higher on collaboration and communication, creative thinking, academic performance. |
| [22] Llorent-Vaquero (2020) | To evaluate the experience of creating serious games with Scratch at a university. | 39 university students. | Spain. | Qualitative methodology with survey and questionnaires. | Students find the tool fun and would recommend it, but do not think they would use it again. Gamification's main advantages include the possibility of immediate feedback, the production and management of key challenges in skills development, fostering motivation and interest in students, and their effectiveness in improving the learning process compared to traditional methods. |
| [23] Mead et al. (2019) | To find out if "Scratch 3" supports the sequencing skills of young adults with learning disabilities (LLDD). | 3 students and a group of young adults between 18 and 24 years old. | England. | Project-based learning. Observation instrument. | Students improve their autonomy, behavior, life skills, and learning. |
| [24] Hoffman et al. (2019) | To understand the benefits of the Mobile CS Principles (Mobile CSP) course that uses App Inventor in teaching computer science programming. | 275 teachers and 6000 students. | USA. | Quantitative methodology with surveys and objective tests. | The use of this course with App Inventor increased creative thinking and motivation in students and provided a solid foundation in computer science. |
| [25] Sheng (2020). | To improve teaching effectiveness of basic programming courses. | 768 students in technology careers. | China. | Mixed quantitative methodology with surveys and qualitative methodology with interviews. | The student body considered it complicated and was not motivated to learn to program. The author created a Scratch learning course and hoped that Scratch would be implemented at the university. |

### 2.6. Identification of the Areas of the Studies

The selected studies worked in different areas of education. The programs were implemented in specific disciplines in different areas of knowledge (Table 3).

**Table 3.** Areas of study of Scratch and/or App Inventor.

| Authors | Intercultural and Multimedia Activities | Engineering | Programming | Psychology | Gamification | Computer Science | Mathematics | Informatic |
|---|---|---|---|---|---|---|---|---|
| [15] | | | | | Yes | | Yes | |
| [16] | | | | | | Yes | | |
| [17] | Yes | | Yes | | | | | |
| [18] | | Yes | | | | | | Yes |
| [19] | | Yes | | | | Yes | | |
| [20] | | Yes | | | | Yes | | |
| [21] | | | | Yes | | | | |
| [22] | | | | | Yes | | | |
| [23] | | | | | | | | |
| [24] | | | | | | Yes | | Yes |
| [25] | | | Yes | | | | | |
| Nº | 1 | 3 | 2 | 1 | 2 | 4 | 1 | 2 |

The university degrees and subjects on which the investigations were focused numbered eight in total. The studies in our selection included four in computer science (36.36%; N = 4), three in engineering (27.27; N = 3), two in programming (18.18%; N = 2), two in computer science (18.18%; N = 2), two in gamification (18.18%; N = 2), one in intercultural and multimedia (9.09%; N = 1), one in mathematics (9.09%; N = 1), and one in psychology (9.09%; N = 1). Computer science and engineering were the fields in which these programming tools were used the most.

### 2.7. Identification of Skills and Competencies

The studies also sought to know if there was an effect of improvement in different skills and abilities (Table 4).

**Table 4.** Skills and competencies.

| Authors | Problem Solving | Motivation | Social Interaction | Critical Thinking | Autonomy | Computational Thinking | Attention | Creative Thinking | Communication |
|---|---|---|---|---|---|---|---|---|---|
| [15] | | Yes | Yes | Yes | Yes | | Yes | | |
| [16] | | Yes | | | | | | | |
| [17] | | | | | | | | | |
| [18] | Yes | Yes | | | | | | | |
| [19] | | Yes | | | Yes | | | | |
| [20] | | Yes | | | | | | | |
| [21] | | Yes | | | Yes | Yes | Yes | | Yes |
| [22] | | | | | | | | | |
| [23] | | | Yes | | Yes | | Yes | | |
| [24] | | Yes | Yes | | | | Yes | Yes | |
| [25] | | Yes | | | | Yes | | Yes | |
| N | 1 | 8 | 3 | 1 | 4 | 2 | 4 | 2 | 1 |

Some competencies and skills coincided in the studies; it was observed that of the 11 selected articles, in 8, the relationship of Scratch and/or App Inventor was found to be correlated with an improvement in motivation (72.72%; N = 8); in 4, it was correlated with that of autonomy (36.36%; N = 4); in 4, it was correlated with an increase in attention (36.36%; N = 4); in 3, it was correlated with social interaction (27.27; N = 3); in 2, it was correlated with creative thinking (18.18%; N = 2); in 2, it was correlated with computational thinking (18.18 %; N = 2); in 1, it was correlated with critical thinking (9.09%; N = 1); in 1, it was correlated with communication (9.09%; N = 1); and in 1, it was correlated with problem solving (9.09%; N = 1). The aspects most enhanced by the programming language were mainly motivation, self-control, and social interaction.

The first research was evaluated by [15], whose objective was to demonstrate that a gamified educational design, using programming in App Inventor to understand the elements of mathematics, brought significant improvements. In the second study [16], the authors wanted to use Scratch to teach programming with a constructionist pedagogy to examine the nature of participation in a computer science class in higher education and whether Scratch motivates and increases the involvement of students. The third study [17] focused on discovering the practices and opinions of students participating in multimedia and cross-cultural activities on whether Scratch promotes programming knowledge acquisition. In the fourth study [18], the authors aimed to assess whether using Scratch with a WEB RS APP for problem-solving exercises improved motivation, reduced teacher intervention, and improved computer programming learning. In the fifth study [19], it was suggested that students of industrial engineering and computer systems who used Scratch, after some time, would become demotivated when faced with programming exercises because they did not meet individual expectations. The authors wondered whether using Scratch with its Caramba extension (which includes an exercise-recommendation system based on characteristics such as taste and complexity) would lead to more meaningful learning and motivation. The sixth study [20] assessed whether using Scratch in computer systems engineering favored meaningful learning in students at a university with a pass rate of 43% and achieved increased academic performance and competence development. The seventh study [21] aimed to achieve, in times of COVID-19, more autonomous teaching of students with psychological problems using Scratch with its integration into the virtual simulated experiential teaching system of psychology. In the eighth study [22], 39 students from the University of Seville created a video game with Scratch; the aim was to describe the creation of educational games and evaluate the benefits of achieving this goal using Scratch. The ninth study [23] focused on university students with learning difficulties and used "Scratch 3" to improve organizational thinking skills, improve communication, make them reflect on their concerns and needs, and thus improve their personal safety and develop their basic coding skills as part of a creative activity. The tenth study [24] focused on a mobile CSP course using App Inventor, which helped to broaden participation in CS (computer science) among male, female, Afro-ethnic, and Latino demographic groups and also provided them with a solid foundation in computer science principles and practices by developing their programming competencies. The eleventh study [25] sought to enhance learning to program using Scratch by creating a Scratch course.

### 2.8. Effectiveness and Main Difficulties in Developing Scratch and App Inventor

The analysis of the results of the programs has confirmed that learning occurs in all the investigations, and almost all of them achieved the effect they were looking for, as shown in Table 5.

**Table 5.** Effectiveness of using Scrach and App Inventor.

| Authors | Effectiveness | Main Difficulties |
|---|---|---|
| [15] | Yes | - |
| [16] | Yes | - |
| [17] | Yes | - |
| [18] | Partly | Although more than 50% of the students consider that Scratch+Ers improves, a significant number felt that it lacked precision in the personalized recommendation of exercises and speed of interaction. |
| [19] | Yes | - |
| [20] | Yes | - |
| [21] | Yes | |
| [22] | Yes | - |
| [23] | Yes | - |
| [24] | Yes | - |
| [25] | Yes | |

The authors of [15] researched the collaboration of 60 students in their second level of engineering and found positive results. The research was intended to demonstrate that the gamified electronic questionnaires made in the MATLAB application (created with App Inventor) used by the experimental group would provide better results than the paper questionnaires used by the control group. The gamified questionnaire stimulated students' critical thinking; encouraged social interaction; and increased engagement to improve learning experiences, motivation, attention, and autonomy. More flexible learning was achieved. It was found that gamified electronic questionnaires improved motivation, critical thinking, autonomy, interaction, and learning assimilation. Thus, university teachers should use gamification with their students.

The study in [16] included 96 students in their first year of computer science and concluded that using Scratch motivates students who are beginning to program. Students expressed positive signs of engagement in learning, and as they used Scratch in their first year of the degree, their interest increased, and they had an easy transition to text-based programming in their second year.

In the study of [17], there was a sample of 113 students selected from three universities, with nine nationalities in the group. This study demonstrated the effectiveness of using Scratch as a tool for students to better understand the management and use of multimedia content through the block-based visual programming language.

The study by [18], with a sample of 64 students in computer science and industrial engineering, showed that if the Scratch programming language is used with the RS Web app, students' programming learning is improved. The computer science students were more critical than the engineering students in this association. A significant number of students indicated that they did not agree with the accuracy of Scratch +ERS in recommending exercises based on their computer programming skills or that the interaction time with the system was fast and should thus be improved. Even so, more than 50% of students found this combination to be a better methodology for learning programming.

The study by [19] with a sample of 64 students showed that the extension of Scratch Caramba increased students' motivation for autonomous learning, which increased the use of the program and academic performance. The pass rate achieved using Caramba was over 52%, 8% higher than the rate achieved during a previous experience using only Scratch and 21% higher than the historical results of teaching without this visual programming language.

Another study [20], with 74 students between 17 and 34 years old, taking computer systems, showed that the use of Scratch improved grades significantly from 10.5% to 47.2%

and that students perceived that it helped them improve in basic programming concepts and were motivated to complete more training with Scratch.

Another study by [21] with 81 psychology students showed that the adaptive-learning mode of psychology based on virtual simulation technology with Scratch could improve public psychology students' learning initiative, communication, autonomy, computational thinking, and knowledge acquisition.

Another study [22] found that Scratch could be useful in different educational contexts; students found it fun and would recommend it, but the percentage of students who would use the tool in the future was low. Scratch is useful for teaching programming and for teaching any other educational subject.

The study of [23] addressed a sample of students with learning difficulties and showed specific results for three students as well as general results for the group. These students handled applications created by high school students in Scratch. The group developed their sequencing skills and other important "life skills" by creating a series of correctly formulated algorithms for their "Keeping Safe" applications. Students' attention spans improved, as did the classroom climate and learner autonomy. High satisfaction with the final learning outcomes was evident.

Another study [24], with 275 teachers and more than 6000 students, in which 29% were women and 32% were minorities, showed that through App Inventor, students created socially useful apps that stimulated their motivation and creative thinking, and achieved significant learning of computer programming principles and practices.

The study of [25], conducted on 768 students in technology careers, found that Scratch facilitates the understanding of programming and improves computational and creative thinking, increasing interest in learning. Furthermore, the results of the experience show the convenience of implementing this programming language in universities.

*2.9. Limitations of the Studies*

Most studies (81.8%) found no limitations in their execution and the desired effect. Only one study using Scratch 3 observed that the program was not fully developed, and the students could not use all the functions that it was supposed to develop.

**3. Discussion**

The ideas and studies analyzed mostly showed that studies on App Inventor and the Scratch programming language have been developed with different results in terms of competency development and academic performance. In this section, we discuss the most relevant findings in relation to the topics described in both the theoretical framework and the objectives of this study.

The main objective of the work was to assess the effect of the use of Scratch and App Inventor on the learning of the programming language and the skills and competencies it develops in university students. This review proved that Scratch and App Inventor help develop specific programming skills. Despite two studies being inconclusive regarding the benefits of using Scratch and App Inventor, most emphasize the positive effects of developing skills and competencies in learning programming. The reviewed studies were published between 2016 and 2022 and reflected the preference for using Scratch and/or the App Inventor programming language in university classrooms from a learning approach to programming and competency stimulation. The number of studies conducted on Scratch was higher than on App Inventor. The dispersion of focus and results makes this study necessary, which has focused its analysis on the lines of work and needs to be covered from the educational point of view, in relation to the benefits of using visual programming languages in educational centers, especially universities.

The school, born at the end of the 19th century and whose objectives were literacy and meeting the needs of industrialization, has given what it can give. These objectives are far removed from today's reality. This has led to questioning the current educational model. We are changing from a model that seeks uniformity to a model that rewards creative

thinking. The student is the protagonist of his learning, and the teacher is a guide who facilitates and accompanies him [26].

We need to use active methodologies that awaken students' interest in the subject. We believe that these languages should be used with university students and that a commitment and involvement of the entire educational community is required if we want to consolidate the learning of programming as fundamental learning. Even so, from the studies reviewed, we found that (a) the study samples were varied and mainly focused on students and their training in Scratch language or App Inventor, (b) there is a lack of a clear and brief method for evaluating the effectiveness of implementing programs with these resources in universities, and (c) all the results obtained in the studies show positive effects for using visual programming languages.

In general, programming with Scratch and/or App Inventor works, generates desirable competencies, and favors meaningful learning in students. The results obtained show this fact; nine of the ten studies found the expected results, although two ([18,22]) only achieved it in part. From the studies presented above, results were extracted that coincide with the study conducted in [10]. Hardly any research studies focused on App Inventor have been found, which is strange since this resource is very attractive to students as they can develop applications on mobile devices. Nowadays, almost everyone has a cell phone, and since it is a ubiquitous medium, we can learn with it at any time and place [27].

Planning and including programming using Scratch and/or App Inventor in different subjects included in the university curriculum would give the program the continuous and integrated character required for any knowledge to be acquired and consolidated. Programs should increasingly use ICT with programming languages such as Scratch and App Inventor to achieve 21st century competencies. The study in [15] demonstrates, as did [28], that using a visual programming language improves mathematical thinking through the creative process of problem solving by developing logic and reasoning as students respond to various forms of feedback.

In two studies we selected ([18,19]), there was a concern about whether students became demotivated in their second year of college programming with Scratch. These studies confirmed that it is crucial to know that an active and learner-centered approach leads to a better understanding of programming concepts and practices in university students and higher motivation; thus, these authors recommend implementing such an approach via Scratch, with extensions that improve program customization, such as ERS or Caramba. Visual programming languages are being taught at universities for technology-related careers, but the study in [19] shows that it is useful for psychology, and [23] demonstrates that Scratch helps students with learning difficulties; moreover, [25] confirms that Scratch improves communication and social interaction in disadvantaged groups.

Studies [15,24] using App Inventor and [21,23] Scratch show that visual programming languages improve students' attention.

Studies [15,16,18–21,24,25] agree that Scratch and App Inventor increase motivation, facts that refute the finding of [22] that students find Scratch fun to use and would recommend it but, curiously, would not consider using it again for their studies.

Other studies [15,19,21,23] also demonstrate an increase in autonomy in students using both visual programming languages. The authors of [29–31] indicated that autonomous learning is a key requirement for educators in the 21st century.

Studies [15,24], using App Inventor and [23] Scratch, show that visual programming languages improve social interaction. Additionally, [30,32], focused on programming activities to encourage social interaction between robots and humans using visual programming languages and tangible programming in programmable robots, highlight the importance of artificial intelligence.

The studies included in our selection find both programming languages positive for learning programming and competence development; [33] also indicated after comparing projects from the gallery of both Apps that, although they have differences, both Scratch

and App Inventor favor computational thinking and that App Inventor has almost three times as many projects as Scratch.

Working and planning students' work by promoting the use of Scratch and App Inventor can significantly improve the acquisition of skills such as autonomy, attention, motivation, critical thinking, creative thinking, computational thinking, communication, problem solving, and social interaction. The general nature of the competencies promoted by using these tools will allow students' transversal development in a university education still based on the master teaching model.

The assumption of challenges such as those we propose will allow universities to assume alternative training models based on development and transversal learning.

## 4. Conclusions

The findings obtained from the analysis of these studies support the general objective proposed in this study. We hoped to find in the design that Scratch and App Inventor are currently used in universities, and showed the scarcity of literary publications about them.

Evidence concerning the articles found shows that Scratch and App Inventor are used for application design, programming, and learning in different disciplines such as mathematics, computer science, engineering, psychology, programming, intercultural and multimedia activities, and gamification in university studies. There is a clear trend to use these applications in areas of scientific knowledge.

Scratch and App Inventor stimulate and develop university students' autonomy, attention, motivation, and computational, critical, and creative thinking. They also encourage social interaction, communication, and problem solving. Since they develop transversal competencies, they should be introduced in all disciplines.

Given Scratch and App Inventor's benefits on students' learning, gamification is considered suitable for university teaching. Gamification is a methodological strategy not often used in higher education. The remaining studies suggest that using these applications could favor the development of learning based on games, which would significantly impact students' motivation.

Few application studies have been identified in the field of university education for using Scratch and App Inventor to develop students' skills.

Programming results have been shown to improve using Scratch and App Inventor. In addition, studies have shown that Scratch and App Inventor improve the competence and use of programming languages. Thus, it is necessary to implement these resources in the university classroom.

## 5. Limitations and Prospects

The main difficulties encountered when carrying out this study mainly centered on the lack of research studies focused on applying Scratch and App Inventor at the university level. This reality shows that using these visual programming languages develops fundamental competencies for developing other learning in university education. The significant number of studies published in non-open access journals has been a drawback due to the difficulties inherent in accessing certain databases or journals. The experiences and studies developed in universities have focused on specific experiences or assumptions whose results are not generalizable, although they have made it possible to identify the main lines of work for learning programming.

**Author Contributions:** Equal contribution. These authors contributed equally to this work. All authors have read and agreed to the published version of the manuscript.

**Funding:** This research received no external funding.

**Data Availability Statement:** Not applicable.

**Conflicts of Interest:** The authors declare no conflict of interest.

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
