# Peer review of "Gamification with Scratch or App Inventor in Higher Education: A Systematic Review"

_futureinternet, doi:10.3390/fi14120374_

Round 1

Reviewer 1 Report

An interesting and specifically focussed systematic review.  The findings highlight a potential gap in current knowledge and understanding and as such the study is a useful contribution that might guide further investigation into the use of Scratch and App Inventor.

There are a few minor spelling/typographical errors that such as the misspelling of Objectives in one of the Table headings.  These should be corrected prior to publication.

Author Response

Dear reviewer, thank you for your comments and support of the manuscript.
We have modified the suggested changes. You can see in blue color the new changes. In red color and marked in yellow the deleted text.
Best regards

Reviewer 2 Report

The paper provides an extensive review of the literature on Scratch and App Inventor usage in Universities. The paper provides interesting insights derived from these studies.

There are some typos so there is a need for careful reading and correction (e.g. Egipt). The text in some of the boxes of Figure 1 are not readable. There are some notions like: "3.1 Subsection" or later "5. Conclusions This section is not mandatory but can be added to the manuscript if the discussion is unusually long or complex." which do not make some sense.

Moreover, the commentary following tables 3 and 4 should focus on the highlights (e.g. the majority of the studies refer to ...) and not repeat the information the reader can see on the table.

Author Response

Dear reviewer, thank you for your comments and your support of the manuscript.
We have modified the suggested changes. You can see in blue color the new changes. In red color and marked in yellow the deleted text.
Best regards

Reviewer 3 Report

The paper conducted a systematic review of gamification and the use of visual programming languages (Scratch and App Inventor) in university institutions. I would suggest the author changes the title of the paper to: 

Gamification with Scratch or App Inventor in Higher education: a systematic review.

The authors have updated the paper with careful revisions. The texts on Fig. 1 look vague. Table 6 is most empty and be removed by using texts to describe.

Author Response

Estimado revisor, gracias por sus comentarios y valiosas recomendaciones.
Hemos abordado todos los cambios que ha sugerido.
Se ha cambiado el título por el sugerido por el revisor.

Hemos completado los textos incompletos de la Figura 1 y eliminado la Tabla 6, insertando los comentarios en el texto del manuscrito.

Reviewer 4 Report

This paper is a literature study investigating the current status of gamification using Scratch and App Inventor in higher education. The following comments must be supplemented before it can be evaluated as an academic research.

First, the abstract does not fit the format and must be rewritten.

Second, English proofreading is required. Currently, the content of the paper is not easy to understand.

Third, it is necessary to change the overall structure of the thesis. I don't think there is a need to describe the 1. introduction and 2. objective separately. I strongly recommend that authors must resturcture of the paper.

Fourth, what is the difference between the research limit betwwen 4.5 and 7?

Fifth, in the 5. discussion, the academic and practical implications of this study should be described.

Author Response

Dear reviewer, thank you for your comments and valuable recommendations.
We have addressed all the changes you have suggested.

We have adapted and rewritten the manuscript abstract.
The English language was thoroughly edited, proofread, and edited by the English editing team of MDPI.
We have structured the manuscript, and removed unnecessary sections and numbering.
The academic and practical implications of this study have been described and the limitations and prospective contribution have been elaborated.
Kind regards

Round 2

Reviewer 4 Report

- Increase the resolution of Figure 1.

- L264~302: Combine short paragraphs into a comprehensive description.

Author Response

Dear reviewer, thank you very much for your comments. We have included figure 1 as an image to improve its resolution. We have unified the paragraphs you suggested.
In addition, the entire text of the article has been edited by MDPI's editing service.
We hope that everything is now correct.
Regards
